# Platinum recycling going green via induced surface potential alteration enabling fast and efficient dissolution

Nejc Hodnik[1,2,*], Claudio Baldizzone[1,3,*], George Polymeros[1], Simon Geiger[1], Jan-Philipp Grote[1], Serhiy Cherevko[1,3], Andrea Mingers[1], Aleksandar Zeradjanin[1,3] & Karl J.J. Mayrhofer[1,3,4]

The recycling of precious metals, for example, platinum, is an essential aspect of sustainability for the modern industry and energy sectors. However, due to its resistance to corrosion, platinum-leaching techniques rely on high reagent consumption and hazardous processes, for example, boiling *aqua regia*; a mixture of concentrated nitric and hydrochloric acid. Here we demonstrate that complete dissolution of metallic platinum can be achieved by induced surface potential alteration, an 'electrode-less' process utilizing alternatively oxidative and reductive gases. This concept for platinum recycling exploits the so-called transient dissolution mechanism, triggered by a repetitive change in platinum surface oxidation state, without using any external electric current or electrodes. The effective performance in non-toxic low-concentrated acid and at room temperature is a strong benefit of this approach, potentially rendering recycling of industrial catalysts, including but not limited to platinum-based systems, more sustainable.

[1] Department of Interface Chemistry and Surface Engineering, Max-Planck-Institut für Eisenforschung GmbH, Max-Planck-Strasse 1, 40237 Düsseldorf, Germany. [2] Department of Catalysis and Chemical Reaction Engineering, National Institute of Chemistry, Hajdrihova 19, 1000 Ljubljana, Slovenia. [3] Department of Electrocatalysis, Helmholtz-Institute Erlangen-Nürnberg for Renewable Energy (IEK-11), Forschungszentrum Jülich, Egerlandstr. 3, 91058 Erlangen, Germany. [4] Department of Chemical and Biological Engineering, Friedrich-Alexander-Universität Erlangen-Nürnberg, Egerlandstr. 3, 91058 Erlangen, Germany. * These authors contributed equally to this work. Correspondence and requests for materials should be addressed to N.H. (email: nejc.hodnik@ki.si).

Platinum, as one of the most active and stable materials, is nowadays largely used as catalyst in automotive catalytic converters for purification of exhaust gases, in petro-chemical industry for production of high octane gasoline and in proton exchange membrane (PEM) fuel cells to produce electricity from hydrogen, just to name a few applications[1–3]. Due to the increase in public and also government awareness of global warming and air pollution, especially the development of later sustainable energy technologies is soaring[4]. Due to a constant increase in demand, it is anticipated that platinum supply will soon not be able to match the needs of the global economy. It is, thus, likely to assume that the Pt supply will become a bottleneck for catalysts production due to its very low availability in natural ores, which are to a large extent found in South Africa[5]. Several states, for example, US, Japan and EU, have already recognized this risk and, consequently, for the past several years are actively involved in the design of new policies, and strategies to secure reliable and unhindered access to so-called critical raw materials (CRM). In this scenario, the chemistry of recycling of platinum and other CRMs from end-of-life products, also referred as urban-mining, will become more important than ever.

There are two types of metallurgical processes for extraction of Pt from oxide ores and end-of-use materials. Although the exact details of many of them are industrial intellectual property, they can be classified as either high-temperature technologies, also referred as pyrometallurgy, or, wet-chemistry-based technologies, that is, hydrometallurgy. The first category consists in melting scraps and separating the metals by weight. Generally, it requires large investments and high-energy consumption[6–8]. On the other hand, the second group is based on relatively cheaper chemical-leaching treatments, for instance, highly acidic solutions of strong oxidizing nature such as hot fuming *aqua regia* or pressurized alkaline cyanide solutions. Third possibility is referred as gas phase volatilization[9,10]. Generally speaking, these state-of-the-art Pt dissolution processes are already quite efficient and employed on the industrial scale[6–8,11–16]. Nevertheless, there is definitely room for improvement, as they suffer from several practical and environmental issues: use of toxic chemicals (lixiviants), release of hazardous gases (HCl vapour, $Cl_2$, $NO_x$, nitrosyl chloride and so on) and leaching residues, high reagents and equipment costs, use of concentrated acids, elevated temperatures, high pressures, consumption of chemicals and pollution of waste water and so on[6–8]. Therefore, a breakthrough in the chemistry of Pt recycling is needed for the whole process to become safer, more economical, sustainable and environmentally friendly. Interestingly, a few recent reports showed novel approaches based on organic solvents[17] and ionic liquids[18] for enhanced Pt dissolution. However, their environmental sustainability, economic viability and real industry compatibility is questionable.

Hydrometallurgical processes are essentially electrochemical in nature[19]. For instance, dissolution by *aqua regia* and, in general, all leaching techniques can be considered as a process analogous to electrochemical corrosion. In brief, mixing highly concentrated nitric and hydrochloric acid induces a homogeneous chemical reaction, where the nitric acid and the chlorine gas evolving in the freshly prepared mixture act as oxidizing agents, while the hydrochloric acid acts as complexing agent and prevents the formation of a passive layer. From an electrochemical point of view, the role of the oxidizing agents is to increase the Pt surface potential above the thermodynamic dissolution potential (ca. 1.2 V versus reversible hydrogen electrode (RHE) for Pt)[20]. Generally, these processes are however only effective at elevated temperatures and with continuous consumption of extremely concentrated acids, as, for example, *aqua regia* loses its activity relatively quickly due to the release of chlorine.

Platinum dissolution has been extensively studied by our group in relation to PEM fuel cell electrocatalysts stability. Although a complete understanding of this complex process has still not been achieved, several valuable insights have been gained recently. On the basis of our electrochemical perspective, in the current study we present a conceptually different chemical approach of dissolving Pt at substantially milder conditions compared with state-of-the-art processes. We utilize only low-concentrated hydrochloric acid (0.3 M, to provide a sufficient pH and presence of chlorides), atmospheric pressure, room temperature, and minimal amounts of gases like ozone and carbon monoxide. Specifically, our approach stands on the following crucial concepts: (i) transient platinum dissolution is an aggressive corrosion process that occurs upon oxidation and reduction of a Pt surface, also referred to as anodic and cathodic Pt dissolution[21,22]. Note that the dissolution at constant electrode potentials above the Pt dissolution onset, which could be compared with soaking in acid, is much lower compared with the transient process[23–25]; (ii) the rate of Pt transient dissolution is accelerated by the presence of chlorides and other halides. We show that, by slowing the process of Pt passivation, chlorides dramatically increase Pt dissolution during anodic and cathodic potential excursions and, at the same time, stabilize Pt ions in the solution[25,26]; and (iii) the presence of dissolved CO gas increases Pt cathodic dissolution due to strong Pt–CO interaction, which physically prevents any potential Pt redeposition[27] that may occur when CO (or some other reducing species such as $H_2$) oxidation potential gets below Pt depostition potential. The electrochemical potential of Pt, that is, the Galvani potential difference between metal surface/electrolyte, can be controlled by exposure to an appropriate gas, either oxidative or reductive in nature, most important, without the use of external potential control (potentiostat)[28]. Thus, the core of our approach is the so-called transient electrochemical dissolution[21] process triggered by repetitive cycling between two gases, leading to an electrode-less-induced surface potential alteration (Fig. 1).

## Results

**On-line electrode potentials and dissolution measurements.** To demonstrate and explore the effectiveness of this new approach, we initially focus on a commercial Pt black catalytic system (transmission electron microscopy images in Supplementary Fig. 1). The study firstly verifies our conceptual framework by monitoring the electrode potentials and dissolution of Pt in the defined system of scanning flow cell (SFC)[21,29]. Second, the accumulated knowledge is transferred to the more application-relevant configuration of a small-scale reactor. We employ a SFC as an advanced three electrode electrochemical cell, in which the flowing electrolyte can be fed to different analytical equipment, for example, Inductively coupled plasma-mass spectrometer (ICP-MS)[21,29] or online electrochemical mass spectrometer (OLEMS)[30,31], to obtain time-resolved information on the dissolved ions and/or evolved gases during electrochemical reactions. Generally, the three electrode set-up would allow controlling the potential of the working electrode, for example, a polycrystalline surface or high-surface catalysts. However, in the present case, it is only employed to monitor the alteration of the surface potential, that is, the open circuit potential (OCP), upon exposure to gases. The OCP is a measure of the metal/solution potential difference and corresponds to the condition of equilibrium when both cathodic and anodic reactions have the same rate.

Figure 2a,b confirms the effect of different gases on the surface potential of Pt. In brief, ozone is effectively oxidizing Pt and adjusts the surface potential to slightly above 1.3 V versus RHE,

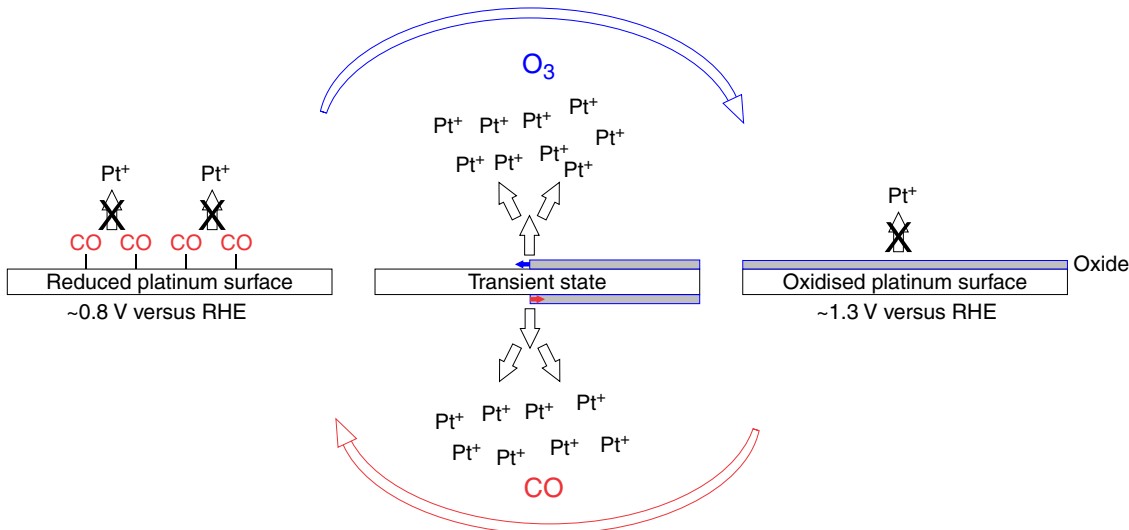

**Figure 1 | Schematic representation of the proposed model for gas-induced transient Pt dissolution.** The transient process of protective Pt-oxide growth induced by $O_3$ purging (blue arrow): this process is relatively slow by nature and also hindered by presence of chlorides[40]; thus, for some critical amount of time unprotected bare Pt surface is exposed to high oxidizing potentials of ozone decomposition (ca. 1.3 V versus RHE) triggering active anodic Pt dissolution[21,25,26]. CO-induced transient process of oxide removal (red arrow): Pt-oxide is reduced by CO triggering Pt cathodic dissolution and again exposing bare Pt to the electrolyte. At stationary conditions (surface on the left and right in the scheme) there is no significant Pt dissolution. At ca. 0.8 V versus RHE Pt is thermodynamically stable and at ca. 1.3 V s. RHE Pt is effectively protected by surface Pt-oxide[20]. Generally, this passivation layer can only be removed at extremely acidic conditions, high chloride concentrations and elevated temperatures—for example, *aqua regia*.

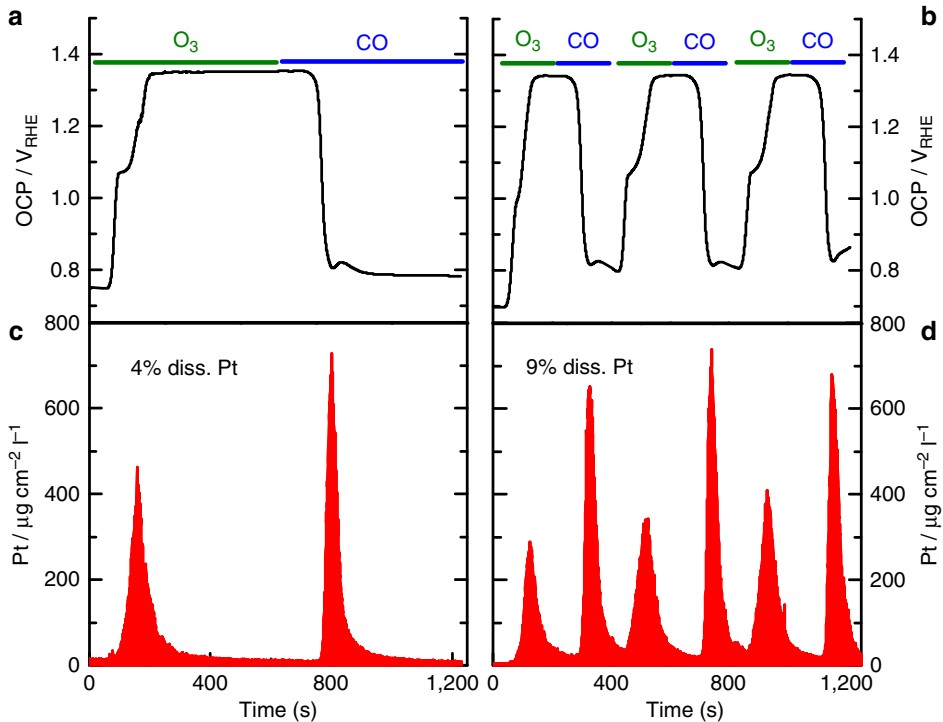

**Figure 2 | Time-resolved Pt gas-induced open circuit potential and subsequent time-resolved Pt dissolution.** (**a**) On-line OCP measurement in one single 10 min ozone and 10 min CO exposure and (**b**) three on-line OCP measurements cycles each containing 3 min ozone and 3 min CO. (**c,d**) present subsequent Pt dissolution profile upon variations in the surface potential (OCP) triggered by ozone and CO exposure in a flow-type electrochemical cell. The analysis also includes the percentage of Pt dissolved in 0.3 M HCl in the same time interval (ca. 20 min).

where ozone reduction and oxygen/chlorine evolution are in equilibrium. In contrast, CO acts as a reductive agent that effectively reduces Pt-oxide and sets the potential to ca. 0.8 V versus RHE, the equilibrium between CO oxidation and reduction of residual oxygen. Note that as the OCP only reaches

the onset potential of the anodic evolution reactions, the amount of hazardous gases ($Cl_2$) evolved is minimal (see OLEMS data in Supplementary Fig. 2). Figure 2c,d clearly demonstrates the predominantly transient nature of Pt dissolution. The peaks of dissolved Pt exactly correspond to the induced shifts in surface

potential, while the dissolution signal quickly drops close to the detection limit upon prolonged exposure to the gases. Figure 2a displays the OCP and Fig. 2c Pt dissolution profile for a cycle of 10 min $O_3$ followed by 10 min CO purging. Figure 2b,d includes the dissolution profile upon three cycles of 3 min $O_3$ and 3 min CO, showing that increasing the frequency of gas exchanges effectively enhances the dissolution yield in the same time interval more than twofolds, that is, from 4% to 9% of the total mass of the catalyst.

Lastly, it is interesting to observe that the extent of the dissolution is not only related to variations in potential, but also the nature of the oxidizing and reducing gases. Supplementary Fig. 3 reports the comparison of the Pt dissolution profiles for 3/3 min $O_3$/CO cycles and the potentiostatically simulated OCP variations, showing a largely enhanced dissolution when the gases are employed. On one hand, the presence of carbon monoxide, as mentioned before, has the additional effect of preventing re-deposition of dissolved species during the surface reduction process. On the other hand, the presence of oxygen radicals related to ozone decomposition most likely have an aggravating effect on dissolution during the oxidation process.

**Complete dissolution measurements.** So far, the dissolution of Pt has been studied in the defined conditions of a three electrode assembly with the catalyst immobilized onto the working electrode in a flow-type electrochemical cell, which allowed us to analyse and understand the underlying fundamental dissolution processes. In a further step, the induced surface potential alteration approach was investigated in a more application-relevant system, that is, a small-scale batch-type reactor (Fig. 4). In this set-up, conversely to the SFC, the Pt black is present in larger quantities (10 mg) and not deposited on any electrode, but rather dispersed in 100 ml of 0.3 M HCl. Moreover, to further increase the amount of chlorides without having to increase the HCl concentration, the solution has been brought to 1 M NaCl (the effect of NaCl molarity on the dissolution yield is reported in Supplementary Fig. 4).

To provide a firm basis on this new system, we first confirmed and optimized our control over the electrochemical potential of the solution by *in operando* monitoring the OCP with a Pt wire electrode. Figure 3 displays the OCP variations induced by a sequence of 5/5 min $O_3$/CO cycles, the optimized protocol for this system. The upper and lower potentials observed for the reactor are in complete agreement with generally expected processes of Pt oxidation and reduction (Supplementary Fig. 5).

Figure 4 demonstrates how effective the procedure in dissolving platinum can be, both visually and quantitatively. Figure 4a includes the concentration of dissolved Pt species in solution versus the number of $O_3$/CO cycles (Supplementary Table 1). Clearly, 20 cycles are more than enough to completely dissolve 10 mg of Pt black. This result is further confirmed by Fig. 4b, showing a distinct evolution of the initial black suspension of metallic Pt black to a yellow-colored solution upon $O_3$/CO cycling, which is a strong indication of the presence of hexachloroplatinic complex. Conversely, exposing the suspension solely to ozone is ineffective at these mild conditions and yields only 0.6% over the same timeframe (Fig. 4a). This is in complete agreement with a previous work by Viñals et al.[32], showing that pure ozone treatment even in 6 M aqueous HCl solution is not strong enough to dissolve Pt. These findings are the direct confirmation of the effectiveness of our novel transient dissolution approach and completely agree with our proposed mechanism in Fig. 1.

## Discussion

Interestingly, a recent work by Lutsuzbaia et al.[33] also utilizes the concept of Pt transient dissolution for recycling platinum from PEM fuel cells. A similar concept, however, with electrochemical pulsed wave methods was already used before[34,35]. Nevertheless, as they apply an external potential by means of a potentiostat, their approach requires electrical contact. Thus, it is limited to conductive catalytic systems deposited on electrodes. Provided that with our system, we easily dissolve state-of-the-art PEM catalysts within few cycles (Supplementary Fig. 6), $O_3$/CO cycling has none of these limitations. As evidence, we have also extended the recycling of platinum and palladium contained in an end-of-life automotive catalytic converter, that is, 0.38 wt.% Pt and 0.24 wt.% Pd on a 99.32 wt.% honeycomb alumina support. Provided some small adjustments in the cycling protocol (Supplementary Fig. 7), Fig. 5 proves that within 35

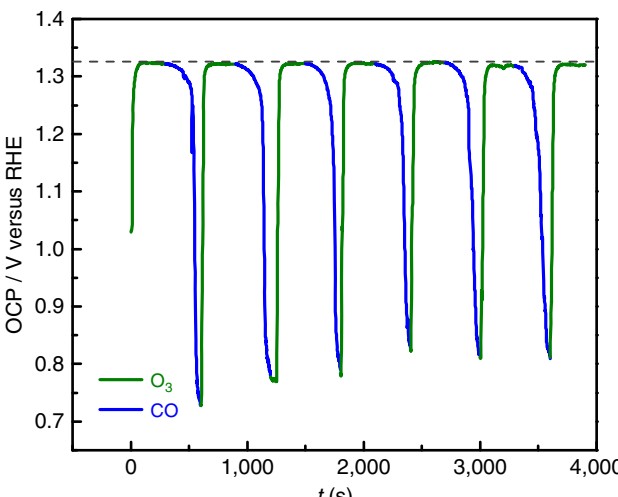

**Figure 3 | Operando open circuit potential measurement within the reactor.** The reactor contained 10 mg Pt black dispersed in a stirred solution of 100 ml 0.3 M HCl and 1 M NaCl.

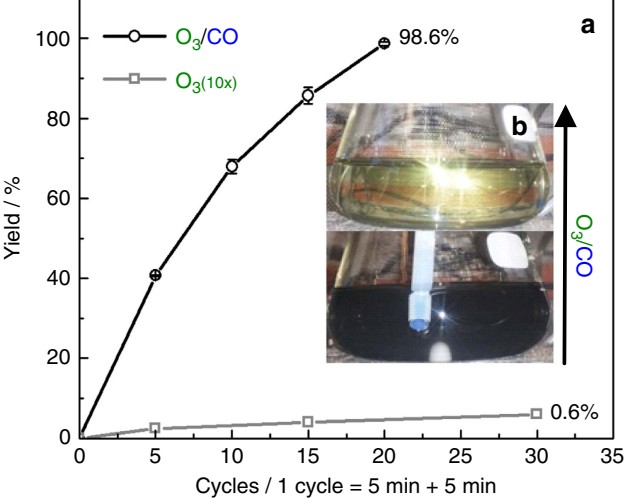

**Figure 4 | Pt dissolution yields. (a)** Pt black dissolution yields in 100 ml of 0.3 M HCl and 1 M NaCl after several cycles of 5 min ozone and 5 min CO purging (black) and after only ozone purging (grey). The dissolution yield was followed by ICP-MS (only ozone) and ICP-OES ($O_3$/CO cycling). The statistical error is s.d. of three experiments. **(b)** Image of the Pt black suspension before (bottom of figure) and after (top of figure) 20 cycles.

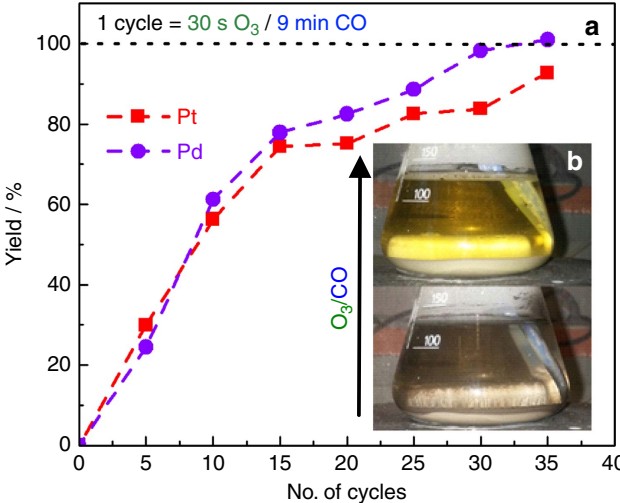

**Figure 5 | Pt and Pd dissolution yields.** (**a**) Pt and Pd dissolution yields from an automotive catalytic converter in 0.5 M HCl and 1 M NaCl versus 35 cycles between 30 s ozone and 9 min CO. Pt and Pd concentrations are measured by ICP-OES. (**b**) Images of the suspension before and after $O_3/CO$ cycling. The car catalyst has been proven to be more resilient to dissolution compared to Pt black probably due to some additional effects related to the alumina support (ca. 99% of the all mass). Specifically, it is known that alumina acts as pH buffer and it can also physically protect the metals.

cycles both metals are completely leached from the non-conductive support.

To conclude, the intention of this study is to present a completely new, environmentally friendly, safe, cheap, scalable and efficient hydrometallurgical concept of platinum leaching/recycling. We utilize the electrochemical transient dissolution mechanism by alternatively employing oxidative and reductive conditions without the use of external potential control (electrodes). The induced surface potential alteration approach opens up a whole new technology to leach and recycle platinum, respectively, from metal ores to end-of-life products at much milder and safer conditions than current state-of-the-art processes. We prove on a fundamental level a complete new concept for Pt chemistry and we pave the way to further engineering optimization like different samples pretreatment, choice of gases (for example, $H_2$ and syngas,), gas flows, pH (alkaline electrolytes), concentrations of chlorides or some other lixiviants (for example, $Br^-$ and $I^-$), hydrodynamic conditions, reactor design, degassing and so on. Finally, as also other noble metals exhibit major transient dissolution[36], we strongly believe the same concept can be applied for the recycling of other CRMs, including the ones out of reach for *aqua regia*, that is, Ru[37,38], Ir[39] and Os[20], which are in addition to Pt also included in our recent German patent application (app. no.: 10 2015 118 279.3).

## Methods

**Scanning flow cell.** The online monitoring of dissolution was achieved by coupling a SFC with an ICP-MS (NexION 300X, Perkin Elmer) with a flow of $180\,\mu l\,min^{-1}$. The catalyst films for analysis with the fully automated SFC system were deposited with a drop-on-demand printer (Nano-PlotterTM 2.0, GeSim) in the form of circular spots of ca. $500\,\mu m$ in diameter onto a glassy carbon substrate. The electrolyte flowing through the SFC is mixed downstream with an internal standard ($^{186}$Re, 10 p.p.b.) in a Y-connector and the resulting electrolyte stream is continuously fed into the ICP-MS, where the dissolved metal ions during the electrochemical treatment are detected online. Coupling of SFC-OLEMS, as a combinatorial technique combines the parameter screening abilities of the SFC

with the possibility for direct detection of volatile products and their correlation to applied potentials or currents. A mass spectrometer with electron impact ionization method is connected to the SFC over a porous Teflon membrane with a pore size of 20 nm that is positioned $50–100\,\mu m$ above the catalysts surface and is therefore in a region of high product concentration. The small distance between the membrane and the electrode, because of that not only results in good sensitivity, but also leads to a fast response time between 1 and 3 s that is mainly determined by the diffusion coefficient of the analysed species. The system allows the simultaneous semi quantitative detection of several products and is therefore well suited for the parallel measurement of oxygen and chlorine.

**Reactor.** A laboratory 250 ml glass beaker with 100 ml of deionized water, 6 g of NaCl (1 M) and 3 ml of 30% HCl (0.3 M). The solution is stirred with a Teflon covered magnetic stirrer (3 cm) at 1,000 r.p.m. The ozone is on-site produced with a concentration of 70 g per $m^3$ of $O_2$ by an ozone generator (Innotec OGVi-8G Lab) and purged with a flow of $1\,l\,min^{-1}$, while carbon monoxide is purged with a flow of $56\,ml\,min^{-1}$. The dissolution yield is measured by sampling and filtering small amounts of the solution at every five-cycle interval. The diluted samples were then fed to either an ICP-MS or ICP-OES depending on the concentration. The OCP was measured by immersing a Pt wire as working electrode, graphite rod as counter electrode and Ag/AgCl as reference electrode and simply tracking the potential by setting current to zero. To avoid interference with the leaching yield results, the *in operando* OCP (Fig. 3) and Pt black dissolution (Fig. 4) measurements have been carried out in different batch reactors under the same identical conditions.

**Data availability.** All relevant data are available from the authors on request.

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

## Acknowledgements

This research was supported by a Marie Curie Intra European Fellowship within the 7th European Community Framework Programme.

## Author contributions

N.H. and C.B. contributed equally to this work. N.H. and C.B. conceived and designed the experiments, and contributed to all other aspects of the research. G.P. and S.G. performed the SFC-ICP-MS experiments. J.-P.G. performed the SFC-OLEMS experiments. A.M. measured platinum and palladium concentrations with ICP-MS and ICP-OES. S.C., A.Z. and K.J.J.M. contributed to the theoretical explanations through discussions and paper writing.
