## [Peer Review File · Nature Communications]

Reviewers' comments:

Reviewer #1 (Remarks to the Author):

This is an interesting paper describing dissolution of Pt under the influence of alternate cycles of O₃ and CO. While a very useful advance, this work appears to build substantially on previously published work by Latsuzbaia et al, ChemSusChem, 2015, 8, 1926 - ref 30 in this MS. The major advance here is that, instead of applying a potential, cycling gases are used to produce oxidizing/reducing conditions, which the authors claim to be greener and more widely applicable as this does not require the Pt to be recovered to be in direct contact with (e.g. immobilized on) an electrode. The former assertion is questionable as O₃ and CO are hardly innocuous gases - it may be correct that this process is greener, but some data would be required to support this. It is certainly correct that this process is more widely applicable and the work should be published, but it is the considered opinion of this referee that this relatively small (albeit very useful) advance would be more appropriately presented in a journal such as ChemSusChem. Finally, the manuscript requires some revision to correct grammatical and spelling errors (e.g. gases, not gasses), but this will be easily done.

Reviewer #2 (Remarks to the Author):

The authors present a new method for the recovery of precious metals (in this example Pt) from end of life materials in acidic chloride media. They utilise a chemical method repeatedly oxidising then reducing the surface with O₃ and CO. This avoids the build up of a passive layer which shuts down dissolution. The acid media used is fairly low concentration and significantly more benign than traditional hydrometallurgical routes.

The case for the importance of Pt recovery is well made and inescapable so the topic is very important. Their use of chemically cycling the potential to dissolve Pt is likely to be new and is consequently of significant interest. There are possibly a few comments going through the text which could be considered.

1) There is a third less well researched method proposed for recovery of PGM which could be cited. It is referred to by authors as 'gas phase volatilisation'. During the 80s and 90s when emissions regs started to bite the auto industry there was a lot of activity on PGM recovery (Pt, Pd and Rh) For example:

Mishra, R.K. Recovery of platinum group metals from automobile catalytic converters - a review. in Precious Metals '89. 1989.

Hoffmann, J.E. Recovery of platinum group metals from automotive catalysts. in Precious and Rare Metal Technologies. 1988.

Woo, S., Kim, C., and Jeon, S., Recovery of platinum-group metals from recycled automotive catalytic converters by carbochlorination. Ind. Eng. Chem. Res., 2000. 39(5): p. 1185-1192.

The last is possibly of interest as it utilises both Cl₂ and CO.

The search for a more benign route to reclaim PGM is a great goal that others have looked into trying to use ammoniacal solutions, urea and the larger halides. From all these I believe that the iodide system has been most successful at allowing dissolution at neutral pH and only modestly oxidising conditions but at elevated temperatures, e.g.

Dawson, R. J. and G. H. Kelsall (2013). "Pt Dissolution and Deposition in High Concentration Aqueous Tri-Iodide/Iodide Solutions." ECS Electrochemistry Letters 2(11): D55-D57.

Though I think there are problems with solubility - so requires careful process design

Zanjani, A. and Baghalha, M., Factors affecting platinum extraction from used reforming catalysts in iodine solutions at temperatures up to 95 {degree sign}C. Hydrometallurgy, 2009. 97(1-2): p. 119-125.

From this it could well be worth looking at the other halides possibly in future?

I am a little unsure of the argument in for CO adsorption preventing Pt redeposition. It would be worth checking the potentials for the CO/CO₂ vs the Pt/PtCl₄²⁻ (aq). At 1.3M activity of chloride and say a dissolved activity of 1e-5M then the Pt/PtCl₄²⁻ reversible potential is about 0.68V (SHE) which I think could be below the CO/CO₂ potential (looks about 0.8(RHE)? so deposition should not occur. The Pt/PtO potential is about 0.92V (SHE) at pH1 so is reducible by the CO as you show. Hence, I am not sure your statement about Pt being stable in chloride at 0.8V (RHE) is correct and there should not be too much concern over Pt redepositing (as metal - if I have understood you correctly)

I like Figure 2 as this shows very succinctly the effective dissolution which is achieved and very nicely the corrosion potential.

It might be good to contrast this more with the electrochemical pulsed wave methods for enhanced Pt dissolution in the text (you give a recent example for PEM): A couple of examples are:

Llopis, J. and A. Sancho (1961). "Electrochemical corrosion of platinum in hydrochloric acid solutions." J. Electrochem. Soc. 108: 720-726.

Benke, G. and W. Gnot (2002). "The electrochemical dissolution of platinum." Hydrometallurgy 64(3): 205-218.

These rely on electronic contact but the principle is the same.

With the autocat did this contain Rh as well? This is traditionally quite a challenge to recover.

Overall it seems an innovative process and a good contender for a more benign recovery route though I believe there are others. It would be good to see further optimisation in a longer paper for the future - other halides, effect of temperature, optimised O₃ and CO (a bit unclear how phases are contacted - small bubbles through a frit? gas dissolves in electrolyte?)

Reviewer #3 (Remarks to the Author):

Hodnik et al. present a very interesting study concerning the recycling (dissolution) of Pt via gas purging. To the reviewers knowledge the described approach is new and innovative and acceptance for publication is thus recommended.

The authors convincingly demonstrate that by a repetitive change from oxidative to reductive conditions (purging by O₃ and CO) Pt catalysts can be efficiently dissolved. Although the reviewer is no expert in industrial precious metal recycling, the described alternative approach for Pt dissolution by aqua regia is performed in the reviewers research group as well. From the presented results the gas purging approach seems to be an interesting alternative. The concept is a nice transfer of knowledge acquired from studying the stability of Pt based catalysts where by the Mayrhofer group and others it has been shown that especially the change from oxidation to reduction of Pt (by the applied electrode potential) is extremely detrimental to catalyst stability. Some minor comments:

Reduction by CO seems for industrial recycling not optimal if large amounts are used. Is there an alternative to CO? Have the authors determined how much gas purging (amount of gas per g Pt) is required for the oxidation and reduction?

Furthermore it would be interesting to discuss more detailed if the effect can be generally applied to precious metals (Pt and Pd is demonstrated).

Reviewers' comments:

Reviewer #1 (Remarks to the Author):

This is an interesting paper describing dissolution of Pt under the influence of alternate cycles of O₃ and CO. While a very useful advance, this work appears to build substantially on previously published work by Latsuzbaia et al, ChemSusChem, 2015, 8, 1926 - ref 30 in this MS.

The major advance here is that, instead of applying a potential, cycling gases are used to produce oxidizing/reducing conditions, which the authors claim to be greener and more widely applicable as this does not require the Pt to be recovered to be in direct contact with (e.g. immobilized on) an electrode. The former assertion is questionable as O₃ and CO are hardly innocuous gases - it may be correct that this process is greener, but some data would be required to support this.

It is certainly correct that this process is more widely applicable and the work should be published, but it is the considered opinion of this referee that this relatively small (albeit very useful) advance would be more appropriately presented in a journal such as ChemSusChem.

Finally, the manuscript requires some revision to correct grammatical and spelling errors (e.g. gases, not gasses), but this will be easily done.

Authors reply:

We thank the reviewer for his careful observation. And we agree with his observation that the concept is similar as in the paper of Latsuzbaia et al, ChemSusChem, 2015, 8, 1926 - ref 30 in this MS. However, besides the fact that our approach has clearly more advantages and is more applicable as it can dissolve metal from uncondusive supports like ceramic catalysts (i.e. catalytic converters, that are one of the major sources of Pt etc.), our breakthrough also opens up a whole new field of study with numerous possible parameters to change and possibilities to leach and, therefore recycle, also other platinum group metals and other CRMs from end-of-life products and possibly also metal ores. Additionally, the innovative concept of controlling the potentials without electrodes gives is in our opinion applicable in other wet-chemistry processes like ex-situ electrocatalysts activation (pre-leaching or pre-reduction). Additionally, as was mentioned by reviewer 2, the concept in Latsuzbaia et al, ChemSusChem, 2015, 8, 1926 - ref 30 in this MS was actually published before, however not on the fuel cell catalysts: Llopis, J. and A. Sancho (1961). "Electrochemical corrosion of platinum in hydrochloric acid solutions." J. Electrochem. Soc. 108: 720-726 and Benke, G. and W. Gnot (2002). "The electrochemical dissolution of platinum." Hydrometallurgy 64(3): 205-218. And of course many articles published by our group on the subject: Topalov AA, et al. Dissolution of Platinum: Limits for the Deployment of Electrochemical Energy Conversion? Angewandte Chemie International Edition 51, 12613-12615 (2012)., Pavlisic A, et al. The influence of chloride impurities on Pt/C fuel cell catalyst corrosion. Chemical Communications 50, 3732-3734 (2014)., ...

Therefore authors believe that the concept of Latsuzbaia et al, ChemSusChem, 2015, 8, 1926 - ref 30 is appropriately acknowledge in MS and that the work presents clear and strong innovative points compare to the existing literature.

We agree CO and O₃ are clearly not innocuous gases. However compared to the concentrations and general amounts of released hazardous gasses in conventional hydrometallurgical and pyrometallurgical methods our process can be considered as relatively mild – green. Please also note that reviewer 2 recognizes this as he states: “The acid media used is fairly low concentration and significantly more benign than traditional hydrometallurgical routes.”. Additionally, when the process will be optimized on the engineering level (our next stage) the entire CO should convert to CO₂ and entire O₃ should convert to O₂ and H₂O during the Pt recycling process. However we agree that a lot of experiments and data still need to be collected the current work is intended to present this new concept to the public. Also CO and O₃ could maybe be replaced by other less dangerous gasses/liquids/salts.

Reviewer #2 (Remarks to the Author):

The authors present a new method for the recovery of precious metals (in this example Pt) from end of life materials in acidic chloride media. They utilise a chemical method repeatedly oxidising then reducing the surface with O₃ and CO. This avoids the build up of a passive layer which shuts down dissolution. The acid media used is fairly low concentration and significantly more benign than traditional hydrometallurgical routes.

The case for the importance of Pt recovery is well made and inescapable so the topic is very important. There use of chemically cycling the potential to dissolve Pt is likely to be new and is consequently of significant interest. There are possibly a few comments going through the text which could be considered.

1) There is a third less well researched method proposed for recovery of PGM which could be cited. It is referred to by authors as 'gas phase volatilisation'. During the 80s and 90s when emissions regs started to bite the auto industry there was a lot of activity on PGM recovery (Pt,Pd and Rh) For example:

Mishra, R.K. Recovery of platinum group metals from automobile catalytic converters - a review. in Precious Metals '89. 1989.

Hoffmann, J.E. Recovery of platinum group metals from automotive catalysts. in Precious and Rare Metal Technologies. 1988.

Woo, S., Kim, C., and Jeon, S., Recovery of platinum-group metals from recycled automotive catalytic converters by carbochlorination. Ind. Eng. Chem. Res., 2000. 39(5): p. 1185-1192.

The last is possibly of interest as it utilises both Cl₂ and CO.

The search for a more benign route to reclaim PGM is great goal that others have looked into trying to use ammonical solutions, urea and the larger halides. From all these I believe that the iodide system has been most successful at allowing dissolution at neutral pH and only modestly oxidising conditions but at elevated temperatures, e.g.

Dawson, R. J. and G. H. Kelsall (2013). "Pt Dissolution and Deposition in High Concentration Aqueous

Tri-Iodide/Iodide Solutions." Ecs Electrochemistry Letters 2(11): D55-D57.

Though I think there are problems with solubility - so requires careful process design

Zanjani, A. and Baghalha, M., Factors affecting platinum extraction from used reforming catalysts in iodine solutions at temperatures up to 95 {degree sign}C. Hydrometallurgy, 2009. 97(1-2): p. 119-125.

Authors reply:

We thank the reviewer for his careful and thorough observation, and appreciate his helpful insights. The manuscript quality has increased with the addition of the proposed literature.

We added the 'gas phase volatilization' to the text as one of the possibilities to recover PGM and cite Hoffmann, J.E. Recovery of platinum group metals from automotive catalysts. in Precious and Rare Metal Technologies. 1988 and Woo, S., Kim, C., and Jeon, S., Recovery of platinum-group metals from recycled automotive catalytic converters by carbochlorination. Ind. Eng. Chem. Res., 2000. 39(5): p. 1185-1192.

Unfortunately we could not find reference Mishra, R.K. Recovery of platinum group metals from automobile catalytic converters - a review. in Precious Metals '89. 1989.

Changes in the text:

Third possibility is refereed as 'gas phase volailisation'.^{9,10}

We added these two references to the text: Dawson, R. J. and G. H. Kelsall (2013). "Pt Dissolution and Deposition in High Concentration Aqueous Tri-Iodide/Iodide Solutions." Ecs Electrochemistry Letters 2(11): D55-D57 and Zanjani, A. and Baghalha, M., Factors affecting platinum extraction from used reforming catalysts in iodine solutions at temperatures up to 95 {degree sign}C. Hydrometallurgy, 2009. 97(1-2): p. 119-125 as reference **XX**

From this it could well be worth looking at the other halides possibly in future?

Authors reply:

In the future we plan to study exactly what reviewer suggested: other halides, their concentrations, temperature, amounts of CO and O₃ (ideally they should be consumed during the recycling process), size of bubbles, other noble metals et cetera.

I am a little unsure of the argument in for CO adsorption preventing Pt redeposition. It would be worth checking the potentials for the CO/CO₂ vs the Pt/PtCl₄²⁻ (aq). At 1.3M activity of chloride and say a dissolved activity of 1e-5M then the Pt/PtCl₄²⁻ reversible potential is about 0.68V (SHE) which I think could be below the CO/CO₂ potential (looks about 0.8(RHE)? so deposition should not occur. The Pt/PtO potential is about 0.92V (SHE) at pH1 so is reducible by the CO as you show. Hence,

I am not sure your statement about Pt being stable in chloride at 0.8V (RHE) is correct and there should not be too much concern over Pt redepositing (as metal - if I have understood you correctly)

I like Figure 2 as this shows very succinctly the effective dissolution which is achieved and very nicely the corrosion potential.

Authors reply:

The reviewers thinking regarding CO is correct for the given case (equilibrium range) and probably for many other. However, there could be a case where Pt/PtCl₄²⁻ (aq) would shift upwards due to Nernst equilibrium (low c(Cl⁻) and high c(Pt/PtCl₄²⁻)) and CO/CO₂ could shift downwards due to presence of oxophilic surface species like Pt defects and Ru (or other oxophilic metals), which is known as a good catalysts for CO (0.5 V RHE). Especially if we consider to use also H₂ and few % of CO. There the CO would definitely help preventing redepositing as was shown in reference Topalov ECOMM.

Further work to examine the effect of redeposition and how CO is influencing it is however planned for the future.

We have included reviewer's the argument in the text.

Changes in the text:

(iii) The presence of dissolved CO gas increases Pt cathodic dissolution due to strong Pt-CO interaction, which physically prevents any potential Pt redeposition that may occur when CO (or some other reducing species like H₂) oxidation potential gets below Pt deposition potential.

It might be good to contrast this more with the electrochemical pulsed wave methods for enhanced Pt dissolution in the text (you give a recent example for PEM): A couple of examples are:

Llopis, J. and A. Sancho (1961). "Electrochemical corrosion of platinum in hydrochloric acid solutions." J. Electrochem. Soc. 108: 720-726.

Benke, G. and W. Gnot (2002). "The electrochemical dissolution of platinum." Hydrometallurgy 64(3): 205-218.

These rely on electronic contact but the principle is the same.

Authors reply:

We have added the two references that utilize electrochemical pulsed wave methods to the text where we discuss electrochemically induced PEM fuel cell electrocatalysts recycling.

Changes in the text:

Similar concept however with electrochemical pulsed wave methods was already used before^{34, 35}.

With the autocat did this contain Rh as well? This is traditionally quite a challenge to recover.

Authors reply:

Our catalysts unfortunately did not contain Rh. However there is a high chance that also Rh would be dissolved since it also exhibits similar transient dissolution, please see references Klemm et. al. Journal of Electroanalytical Chemistry 677–680 (2012) 50–55 and Cherevko et. al. ChemCatChem 6 (2014) 2219–2223. This is also part of our work currently.

Overall it seems an innovative process and a good contender for a more benign recovery route though I believe there are others. It would be good to see further optimisation in a longer paper for the future - other halides, effect of temperature, optimised O₃ and CO (a bit unclear how phases are contacted - small bubbles through a frit? gas dissolves in electrolyte?)

Authors reply:

We thank reviewer for positive and constructive review. The authors surely intend to investigate the abovementioned parameters. Moreover, the authors agree that further optimization is needed, especially, concerning the contacting systems, which in the submitted paper was fairly simple and consisted of a Teflon porous frit or direct bubbling through an open tube.

Reviewer #3 (Remarks to the Author):

Hodnik et al. present a very interesting study concerning the recycling (dissolution) of Pt via gas purging. To the reviewers knowledge the described approach is new and innovative and acceptance for publication is thus recommended.

The authors convincingly demonstrate that by a repetitive change from oxidative to reductive conditions (purging by O₃ and CO) Pt catalysts can be efficiently dissolved. Although the reviewer is no expert in industrial precious metal recycling, the described alternative approach for Pt dissolution by aqua regia is performed in the reviewers research group as well. From the presented results the gas purging approach seems to be an interesting alternative. The concept is a nice transfer of knowledge acquired from studying the stability of Pt based catalysts where by the Mayrhofer group and others it has been shown that especially the change from oxidation to reduction of Pt (by the applied electrode potential) is extremely detrimental to catalyst stability.

Some minor comments:

Reduction by CO seems for industrial recycling not optimal if large amounts are used. Is there an alternative to CO? Have the authors determined how much gas purging (amount of gas per g Pt) is required for the oxidation and reduction?

Furthermore it would be interesting to discuss more detailed if the effect can be generally applied to precious metals (Pt and Pd is demonstrated).

Authors reply:

We thank the reviewer for his positive review and interesting comments.

One option for replacing CO would be to use methanol oxidation where also CO is the most “stubborn” intermediate. The other option would be to use hydrogen with the addition of CO in few %. However ideally after the engineering optimization the entire CO would be anyway transformed

to CO₂ and no harmful species would be released. Nevertheless, further studies are necessary to make this process more efficient (measure CO and O₃ consumptions) and especially greener.

We strongly believe that indeed this process can be used for other noble and precious metals as since they all exhibit transient dissolution behavior when electrochemically cycled: Cherevko et. al. ChemCatChem 6 (2014) 2219–2223 and other our publications...

REVIEWERS' COMMENTS:

Reviewer #2 (Remarks to the Author):

Thank you for the clarifications and additions. This is a valuable and novel piece of work and I look forward to reading further of your developments.